# Streaming Partitioning of RDF Graphs for Datalog Reasoning

Temitope Ajileye[0000−0002−3657−7624], Boris Motik[0000−0003−2506−4118], and Ian Horrocks[0000−0002−2685−7462]

Department of Computer Science
University of Oxford
Oxford, United Kingdom

**Abstract.** A cluster of servers is often used to reason over RDF graphs whose size exceeds the capacity of a single server. While many distributed approaches to reasoning have been proposed, the problem of data partitioning has received little attention thus far. In practice, data is usually partitioned by a variant of hashing, which is very simple, but it does not pay attention to data locality. Locality-aware partitioning approaches have been considered, but they usually process the entire dataset on a single server. In this paper, we present two new RDF partitioning strategies. Both are inspired by recent *streaming* graph partitioning algorithms, which partition a graph while keeping only a small subset of the graph in memory. We have evaluated our approaches empirically against hash and min-cut partitioning. Our results suggest that our approaches can significantly improve reasoning performance, but without unrealistic demands on the memory of the servers used for partitioning.

## 1 Introduction

The *Resource Description Framework* (RDF) is a popular data format, where *triples* represent relationships between *resources*. The *Web Ontology Language* (OWL) is layered on top of RDF to structure the data and support *reasoning*: a reasoner can derive fresh triples using domain knowledge. Thus, developing efficient reasoning algorithms for RDF has received considerable attention.

A popular way to realise OWL reasoning is to encode the rules of inference in a prominent rule-based formalism called *datalog*. For example, the OWL 2 RL profile is a fragment of OWL designed to support datalog reasoning. Datalog reasoning is often implemented in practice by *materialisation*: all consequences of the data and the rules are precomputed in a preprocessing step so that queries can later be evaluated without any further processing of the rules.

Modern RDF datasets can be very large; for example, the UniProt[1] dataset contains over 34 billion triples. Complex reasoning over such large datasets is infeasible on a single server, so a common solution is to partition the data in a cluster of shared-nothing servers. Many such approaches for RDF querying have been

---

[1] https://www.uniprot.org/

proposed [13, 23, 11, 30, 12, 16, 28, 4, 10, 22]. Reasoning is more involved since it requires interleaving queries and updates, but nevertheless several distributed RDF reasoners have been developed [24, 27, 26, 9, 17, 29].

Rule application during reasoning requires distributed join processing, which can be costly if the triples to be joined are stored in different servers; moreover, derived triples need to be distributed across the cluster. Thus, data should ideally be partitioned in a locality-aware way to minimise overheads. Little attention has been paid thus far to the data partitioning problem. Systems based on Hadoop and Spark store the data in a distributed file system and thus typically cannot influence data placement. Systems that explicitly control data placement usually determine a triple's destination by hashing some or all of the triple's components (usually the subject). Hashing is very simple to implement and requires little resources, but it can incur significant overhead, particularly for subject–object and object–object joins. Other systems use min-cut graph partitioning [15] to obtain locality-aware partitions; however, this usually requires loading all data into a single server, which defeats the main goals of using a cluster.

*Streaming* methods aim to produce good graph partitions without loading the entire graph into memory at any point in time (but by possibly reading the graph data several times). Such techniques have been developed primarily for general graphs, rather than RDF. Motivated by the desire to improve the performance of distributed RDF reasoners, in this paper we adapt the HDRF [21] and 2PS [19] state-of-the-art streaming graph partitioning algorithms to RDF. Unlike HDRF and 2PS, our $HDRF_3$ and $2PS_3$ algorithms have to take into account certain idiosyncrasies of the RDF data model. For example, it is well known that subject–subject joins are very common in RDF queries, so colocating triples with the same subject is really important in RDF; however, honouring this requires modifications to HDRF and 2PS.

By comparing our approaches empirically with hash and min-cut partitioning, we investigated how different data partitioning strategies affect reasoning times and network communication. We based our evaluation on the DMAT distributed datalog reasoner [3]. The reasoning algorithm of DMAT is unique in that it is independent of any specific data partitioning strategy: as long as a certain index is provided that informs the system of how data is distributed in the cluster, the algorithm can correctly compute the materialisation.

We show empirically that partitioning the data into highly connected subsets can be very effective at reducing communication and thus reducing reasoning times; however, it can also lead to workload imbalances among servers, which can lead to increases in reasoning when the communication overhead is small. Overall, our $2PS_3$ method seems to be very effective: while requiring only modest resources for partitioning, it can more than halve the reasoning times compared to hash partitioning. Thus, we believe our technique provides an important building block of truly scalable distributed RDF reasoners.

The proofs of our results, all datasets and rule sets used for testing, and the DMAT system are available as online supplementary material.[2]

---

[2] https://krr-nas.cs.ox.ac.uk/2021/stream-graph-partitioning/

## 2    Preliminaries

We next recapitulate some common definitions. An *RDF graph* $G$ is a finite set of triples of the form $\langle s, p, o \rangle$, where $s$, $p$, and $o$ are *resources* (i.e., IRIs, blank nodes, or literals) called *subject*, *predicate*, and *object*, respectively. The *vocabulary* of $G$ is the set of all resources occurring in $G$. Given a resource $r$, let $G^+(r) = \{\langle s, p, o \rangle \in G \mid s = r\}$ and $G(r) = \{\langle s, p, o \rangle \in G \mid s = r \text{ or } o = r\}$. We call $|G^+(r)|$ and $|G(r)|$ the *out-degree* and the *degree* of $r$, respectively.

A *partition* $\mathcal{P}$ of an RDF graph $G$ is a list of RDF graphs $\mathcal{P} = G_1, \ldots, G_n$ such that $G_i \cap G_j = \emptyset$ for $1 \leq i < j \leq n$ and $G = \bigcup_{i=1}^{n} G_i$. We call graphs $G_i$ *partition elements*. The *replication set* of a resource $r$ is $A(r) = \{k \mid G_k \cap G(r) \neq \emptyset\}$. For $V$ the vocabulary of $G$, the *replication factor* of a partition $\mathcal{P}$ is defined as

$$\mathsf{RF}(G, \mathcal{P}) = \frac{1}{|V|} \sum_{r \in V} |A(r)|.$$

Given a fixed tolerance parameter $\alpha \geq 1$, the objective of graph partitioning is to compute a partition $\mathcal{P}$ of $G$ such that $|G_i| \leq \alpha \frac{|G|}{n}$ holds for each $1 \leq i \leq n$, while minimising the replication factor $\mathsf{RF}(G, \mathcal{P})$. Thus, each $G_i$ should hold roughly the same number of triples, while ensuring that resources are replicated as little as possible. Solving this problem exactly is computationally hard, so the objective is usually weakened in practice. The algorithms we present in this paper will honour the restrictions on the sizes of $G_i$; moreover, they will aim to make the replication factor small, but without minimality guarantees.

A *datalog* rule is an expression of the form $H \leftarrow B_1, \ldots, B_n$, where $H$ and $B_i$ are *atoms* of the form $\langle t_s, t_p, t_o \rangle$, and $t_s$, $t_p$, and $t_o$ are variables or resources. Atom $H$ is called the *head*, and $B_1, \ldots, B_n$ are called the rule *body*. A *substitution* $\sigma$ is a mapping of variable to resources, and $A\sigma$ denotes the result of replacing each variable in atom $x$ with $\sigma(x)$. A rule is applied to an RDF graph $G$ by enumerating each substitution $\sigma$ such that $\{B_1\sigma, \ldots, B_n\sigma\} \subseteq G$, and then extending $G$ with $H\sigma$. To compute the *materialisation* of $G$ for a set of datalog rules $P$, this process is iteratively repeated for each rule $r \in P$ as long as possible—that is, until no new triples can be derived. In this work, we study how different partitioning strategies affect the performance of computing the materialisation when the RDF data is partitioned across a cluster of servers.

## 3    Related Work

In this section, we present an overview of the related approaches to distributed querying, distributed reasoning, and RDF data partitioning.

**Distributed Query Processing.** To compute a join in a distributed setting, facts participating in the join must be brought to a server in the cluster. Many solutions to this key technical problem have been developed. Numerous systems (e.g., HadoopRDF [13] and S2RDF [23], to name a few) are built on top of big data frameworks such as Hadoop or Spark. Systems such as YARS2 [11] and

Trinity.RDF [30] compute joins on a single server after retrieving data from the cluster. Systems such as H-RDF-3X [12], SHAPE [16], and SemStore [28] split a query into parts that can be evaluated without communication, and then combine the partial answers in a final join phase. Finally, systems such as AdPart [4] and TriAd [10] compute distributed joins by exchanging partial answers between servers. Recently, 22 systems were surveyed and 12 of those were compared experimentally [1], and TriAd and AdPart were identified as fastest. The *dynamic data exchange* [22] approach was later shown to be also very competitive.

**Distributed Reasoning.** Matching rule bodies corresponds to query evaluation, so distributed reasoning includes distributed querying; however, it also involves distributed data updates, which introduces additional complexity. SociaLite [24] handles datalog extended with a variant of monotonic aggregation. Many distributed RDF reasoners can handle only limited datalog subsets [5]. For example, RDFS reasoning can be performed without any communication [27]. WebPIE [26] handles the OWL-Horst fragment using Hadoop, while CiChild [9] and SPOWL [17] handle the OWL-Horst and the OWL 2 RL fragments, respectively, in Spark. PLogSpark [29], also implemented in Spark, is one of the few distributed RDF reasoners that can handle arbitrary datalog rules.

**DMAT.** Our DMAT system [3] supports distributed seminaïve evaluation of arbitrary datalog rules by extending the distributed query answering technique by Potter et al. [22]; the system uses RDFox [20] for triple storage, indexing, and retrieval. DMAT uses an index to locate the relevant data in the cluster, allowing it to be used with any partitioning strategy. This is different from most existing approaches, where the reasoning algorithms depend on the details of data partitioning. We use DMAT in our evaluation since it allows us to vary the partitioning strategies only and study how this affects the performance of reasoning. While the absolute reasoning times are specific to DMAT, the number of joins that span servers are the same for all implementations, so other systems should exhibit similar relative performance for different partitioning strategies.

**Data Partitioning.** Although it is intuitive to expect that partitioning the data carefully to minimise communication should improve the performance of distributed systems, the effects of data partitioning remain largely unknown. Existing approaches to data partitioning can be broadly divided into three groups. The first groups consists of systems that use Hadoop or Spark to store their data in a distributed file system. The data is usually allocated randomly to servers, which makes exploiting data locality during querying/reasoning difficult. The second group consists of hash-based variants, where the destination for a triple is determined by hashing one or more triple's components (usually subject). The third group consists of variants based on min-cut graph partitioning [15], which aims to minimise the number of edges between partitions and thus reduce the cost of communication. Such approaches are sometimes combined with data replication (e.g., [12, 10]), where a triple is stored on more than one server. All systems in the latter two groups colocate triples with the same subjects to eliminate communication for the most common subject–subject joins [8].

# 4   Motivation and Our Contribution

Distributed reasoning requires network communication for evaluating rule bodies and for distributing the derived triples, and communication is much slower than RAM access. One can thus intuitively expect that communication will critically determine the performance of reasoning, and that, to reduce communication and thus improve performance, joining triples are colocated whenever possible.

Janke et al. [14] studied this problem for distributed query processing. Interestingly, they concluded that reducing communication can be detrimental if done at the expense of uneven server workload. However, it is unclear to what extent this study applies to reasoning. Reasoning over large datasets involves evaluating millions of queries and distributing derived triples, both of which can incur much more communication than for evaluating a single query. Moreover, imbalances in single queries could even out over all queries.

Another question is how to effectively partition RDF data in a locality-aware way. As we mentioned in Section 3, subject hashing is commonly used in practice; while very efficient, it does not take the structure of an RDF graph into account and thus provides no locality guarantees for subject–object or object–object joins. Other commonly used approaches are based on min-cut partitioning. The METIS partitioner requires loading the entire graph into a single server, which is clearly problematical. This problem can be mitigated by using the parallelised version of METIS called ParMETIS; however, graph partitioning is an NP-hard problem, so such a solution is still likely to use considerable resources.

Thus, how to partition RDF data effectively, and how this affects distributed reasoning, is still largely unknown. To answer the former question, we draw inspiration from recent work on *streaming graph partitioning* [25, 21, 31, 6, 2, 18, 19] methods, which process the graph edges a fixed number of times without ever storing the entire graph in memory. The memory used by these algorithms is often proportional to the number of graph vertices, which is usually at least an order of magnitude smaller than the number of edges.

These approaches seem to provide a good basis for RDF partitioning, but they are are typically formulated for general (directed or undirected) graphs. Several RDF-specific issues must be taken into account to obtain adequate partitions in the context of RDF. For example, colocating triples with the same subject was shown to be crucial for practical applications (cf. Section 3). Thus, in Sections 5 and 6, we present two new streaming RDF partitioning techniques, which we obtain from the state-of-the-art algorithms HDRF [21] and 2PS [19]. The idea behind the former is to prefer replicating vertices of higher degree so that a smaller number of vertices has to be replicated overall, and the idea behind the latter is to assign to each server communities of highly connected vertices.

In Section 7 we empirically investigate the connection between data partitioning and reasoning performance. To this end, we compare the performance of reasoning for different data partitioning strategies: our two new techniques, subject hash partitioning, and a variant of min-cut partitioning [22]. Our results suggest that data partitioning can indeed have a significant impact on reasoning performance, sometimes cutting the reasoning times to less than half.

## 5    The HDRF$_3$ Algorithm

We now present our HDRF$_3$ algorithm for streaming partitioning of RDF data. We follow the 'high degree replicated first' principle from the HDRF algorithm for general graphs [21]. In Section 5.1 we briefly discuss the original idea, and in Section 5.2 we discuss in detail how we adapted these principles to RDF.

### 5.1    High Degree Replicated First Streaming Partitioning

The HDRF algorithm [21] targets large undirected graphs whose vertex degree distribution resembles the power-law distribution. The algorithm aims to replicate (i.e., assign to more than one server) vertices with higher degrees, so that a smaller number of vertices is replicated overall. It processes sequentially the edges of the input graph and assigns them to servers. For each server $k \in \{1, \ldots, n\}$, the algorithm maintains the number $N_k$ of eges currently assigned to server $k$; all $N_k$ are initially zero. For each vertex $v$, the algorithm maintains the degree $deg(v)$ of $v$ in the subgraph processed thus far, and the replication set $A(v)$ for $v$. For each $v$, the degree $deg(v)$ is initialised to zero, and $A(v)$ is initialised to the empty set. To allocate an undirected edge $\{v, w\}$, the algorithm first increments $deg(v)$ and $deg(w)$, and then for each candidate server $k \in \{1, \ldots, n\}$ it computes the score $C(v, w, k)$. The algorithm sends the edge $\{v, w\}$ to the server $k$ with the highest score $C(v, w, k)$, and it increments $N_k$.

The score $C(v, w, k)$ consists of two parts. The first one estimates the impact that placing $\{v, w\}$ on server $k$ will have on replication, and it is computed as

$$C_{REP}(v, w, k) = g(v, w, k) + g(w, v, k), \qquad \text{where}$$

$$g(v, w, k) = \begin{cases} 1 + \frac{deg(w)}{deg(v) + deg(w)} & \text{if } k \in A(v), \\ 0 & \text{otherwise.} \end{cases}$$

To understand the intuition behind this formula, assume that vertex $v$ occurs only on server $k$, vertex $w$ occurs only server $k'$, and $deg(v) > deg(w)$. Then, we have $g(v, w, k) < g(w, v, k')$, which ensures that edge $\{v, w\}$ is sent to server $k'$—that is, vertex $v$ is replicated to server $k'$, in line with our desire to replicate higher-degree vertices. The sum $deg(v) + deg(w)$ in the denominator of the formula for $g(v, w, k)$ is used to normalise the degrees of $v$ and $w$.

Considering $C_{REP}(v, w, k)$ only would risk producing partitions of unbalanced sizes. Therefore, the second part of the score is used to favour assigning edge $\{v, w\}$ to the currently least loaded server using formula

$$C_{BAL}(k) = \frac{maxsize - N_k}{\epsilon + maxsize - minsize},$$

where $maxsize$ and $minsize$ are the maximal and minimal possible partition sizes.

Scores $C_{REP}(v, w, k)$ and $C_{BAL}(k)$ are finally combined using a fixed weighting factor $\lambda$ as

$$C(v, w, k) = C_{REP}(v, w, k) + \lambda \cdot C_{BAL}(k)$$

By tuning $\lambda$, we can determine how important is minimising imbalance in partition sizes as opposed to achieving low replication factors.

The version of the algorithm presented above makes just one pass over the graph edges, and $g(v, w, k)$ and $g(w, v, k)$ are computed using the partial vertex degrees (i.e., degrees in the subset of the graph processed thus far). The authors of HDRF also discuss a variant where exact degrees are computed in a preprocessing pass. The authors also show empirically that this does not substantially alter the quality of the partitions that the algorithm produces.

## 5.2   Adapting the Algorithm to RDF Graphs

Several problems need to be addressed to adapt HDRF to RDF graphs. A minor issue is that RDF triples correspond to labelled directed edges, which we address by simply ignoring the predicate component of triples. A more important problem is to ensure that all triples with the same subject are colocated on a single server, which, as we already mentioned in Section 4, is key to ensuring good performance of distributed RDF systems. To address this, we compute the destination for all triples with subject $s$ the first time we see such a triple.

The pseudo-code of HDRF$_3$ is shown in Algorithm 1. It takes as input a parameter $\alpha$ determining the maximal acceptable imbalance in partition element sizes, the balance parameter $\lambda$ as in HDRF, and another parameter $\delta$ that we describe shortly. In a preprocessing pass over $G$ (not shown in the pseudo-code), the algorithm determines the size of the graph $|G|$, and the out-degree $|G^+(r)|$ and the degree $|G(r)|$ of each resource $r$ in $G$. The algorithm also maintains (i) the replication set $A(r)$ for each resource, which is initially empty, (ii) a mapping $T$ of resources occurring in subject position to servers, which is initially undefined on all resources, and (iii) the numbers $N_1, \ldots, N_n$ and $R_1, \ldots, R_n$ of triples and resources, respectively, assigned to servers thus far, which are initially zero.

The algorithm makes a single pass over the graph and processes each triple $\langle s, p, o \rangle \in G$ using the function PROCESSTRIPLE. Mapping $T$ keeps track of the servers that will receive triples with a particular subject resource. Thus, if $T(s)$ is undefined (line 2), the algorithm sets $T(s)$ to the server with the highest score (line 3) in a way analogous to HDRF. All triples with the same subject encountered later will be assigned to server $T(s)$, so counter $N_{T(s)}$ is updated with the out-degree of $s$ (line 4). Finally, the triple is sent to server $T(s)$ (line 5), and the replication sets of $s$ and $o$ and the number of resources $R_{T(s)}$ on server $T(s)$ are updated if needed (lines 6 and 7).

The score of sending triple $\langle s, p, o \rangle$ to server $k$ is calculated as in HDRF. The replication part $C_{REP}$ of the score is computed in lines 11 and 13. Unlike the original HDRF algorithm, we assign all triples with subject $s$ to a server the first time we encounter resource $s$, so having complete degree is important to take into account the impact of further triples with the same subject. Moreover, we observed empirically that it is beneficial for the performance of reasoning to have partition elements with roughly similar average resource degrees. Function DEG estimates the current average degree of resources in server $k$ as a quotient of the currently numbers of triples ($N_k$) and resources ($R_k$) assigned to server

---

**Algorithm 1** $\mathrm{HDRF}_3$

---

**Require:** tolerance parameter $\alpha > 1$
             the balance parameter $\lambda$
             the degree imbalance parameter $\delta$
             the target number of servers $n$
             $|G|$, $|G^+(r)|$, and $|G(r)|$ for each resource $r$ in $G$ are known
             $A(r) \coloneqq \emptyset$ for each resource $r$ in $G$
             Mapping $T$ of resources to servers, initially undefined on all resources
             $N_k \coloneqq R_k \coloneqq 0$ for each server $k \in \{1, \ldots, n\}$

1:  **function** $\mathrm{PROCESSTRIPLE}(s, p, o)$
2:     **if** $T(s)$ is undefined **then**
3:         $T(s) \coloneqq \arg\max_{k \in \{1, \ldots, n\}} \mathrm{SCORE}(s, o, k)$
4:         $N_{T(s)} \coloneqq N_{T(s)} + |G^+(s)|$
5:     Add $(s, p, o)$ to $G_{T(s)}$
6:     **if** $T(s) \notin A(s)$ **then** Add $T(s)$ to $A(s)$ and increment $R_{T(s)}$
7:     **if** $T(s) \notin A(o)$ **then** Add $T(s)$ to $A(o)$ and increment $R_{T(s)}$

8:  **function** $\mathrm{SCORE}(s, o, k)$
9:     $C_{REP} \coloneqq 0$
10:    **if** $k \in A(s)$ and $\mathrm{DEG}(k) \leq \min_{\ell \in \{1, \ldots, n\}} \mathrm{DEG}(\ell) + \delta$ **then**
11:       $C_{REP} \coloneqq C_{REP} + 1 + \frac{|G(o)|}{|G(s)| + |G(o)|}$
12:    **if** $k \in A(o)$ and $\mathrm{DEG}(k) \leq \min_{\ell \in \{1, \ldots, n\}} \mathrm{DEG}(\ell) + \delta$ **then**
13:       $C_{REP} \coloneqq C_{REP} + 1 + \frac{|G(s)|}{|G(s)| + |G(o)|}$
14:    $C_{BAL} \coloneqq 1 - n\frac{N_{k'} + |G^+(s)|}{\alpha|G|}$
15:    **return** $C_{REP} + \lambda\frac{\sum_k N_k}{|G|} C_{BAL}$

16: **function** $\mathrm{DEG}(k)$
17:    **return** $(R_k = 0)\ ?\ 0\ :\ N_k / R_k$

---

$k$. Then, in lines 11 and 13, $C_{REP}$ is updated only if the average degree of server $k$ is close (i.e., within $\delta$) to the minimal average degree.

The balance factor is computed in line 14, and it is obtained by taking into account that the maximum size of a partition element is $\alpha|G|/n$.

Finally, $C_{REP}$ and $C_{BAL}$ are combined using $\lambda$ in line 15. However, unlike the original HDRF algorithm, factor $\frac{\sum_k N_k}{|G|}$ ensures that partition balance grows in importance towards the end of partitioning.

As we mentioned in Section 2, producing a balanced partition while minimising the replication factor is computationally hard, so the minimality requirement is typically dropped. The following result shows that Algorithm 1 honours the balance requirements, provided that $\alpha$ and $\lambda$ are chosen in a particular way.

**Proposition 1.** *Algorithm 1 produces a partition that satisfies $|G_i| \leq \alpha \frac{|G|}{n}$ for each $1 \leq i \leq n$ whenever $\alpha$ and $\lambda$ are selected such that*

$$\alpha > 1 + n\frac{\max_r |G^+(r)|}{|G|} \quad and \quad \lambda \geq \frac{4\alpha}{n\left(\frac{\alpha-1}{n} - \frac{\max_r |G^+(r)|}{|G|}\right)^2}.$$

## 6   The 2PS₃ Algorithm

We now present our 2PS₃ algorithm for RDF, which adapts the *two-phase streaming* algorithm 2PS [19]. In Section 6.1 we discuss the original idea, and in Section 6.2 we discuss in detail how to apply these principles to RDF.

### 6.1   Two-Phase Streaming

The 2PS algorithm processes undirected graphs in two phases. In the first phase, the algorithm clusters resources into communities with the goal of placing highly connected resources into a single community. This is achieved by initially assigning each resource in the graph to a separate community. Then, when processing an edge $\{v, w\}$ in the first phase, the current sizes of the current communities of $v$ and $w$ are compared, and the resource belonging to the smaller community is merged into the larger community. Thus, communities are iteratively coarsened as edges of the input graph are processed in the first phase. The entire first phase can be repeated several times to improve community detection.

After all edges are processed in the first phase, the identified communities are greedily assigned to servers. Then, the graph is processed in the second phase, and edges are assigned to the communities of their vertices.

### 6.2   The Algorithm

Just like in the case of HDRF, the main challenge in extending 2PS to RDF is to deal with the directed nature of RDF triples, and to ensure that triples with the same subject are assigned to the same server.

The pseudo-code of 2PS₃ is shown in Algorithm 2. As in HDRF₃, the algorithm uses a preprocessing phase to determine the size of graph $|G|$ and the out-degree $|G^+(r)|$ of each resource. Thus, 2PS₃ uses three phases; however, to stress the relationship with the 2PS algorithm, we call the algorithm 2PS₃.

The algorithm maintains a global mapping $C$ of resources to communities—that is, $C(r)$ is the community of each resource $r$. Thus, two resources $r_1$ and $r_2$ are in the same community if $C(r_1) = C(r_2)$. Initially, each resource $r$ is placed into its own community $c_r$. As the algorithm progresses, the image of $C$ will contain fewer and fewer communities. Once communities are assigned to servers, a triple $\langle s, p, o \rangle$ will be assigned to the server of community $C(s)$, thus ensuring that all triples with the same subject are colocated.

The algorithm also maintains a global function that maps each community $c$ to its size $S(c)$. Please note that $S(c)$ does not hold the number of resources

---

**Algorithm 2** 2PS$_3$

---

**Require:** tolerance parameter $\alpha > 1$
the target number of servers $n$
$|G|$ and $|G^+(r)|$ for each resource $r$ in $G$ are known
$C(r) \coloneqq c_r$ and $S(c_r) \coloneqq |G^+(r)|$ for each resource $r$ in $G$, where
$c_r$ is a community unique for $r$

18: **function** ProcessTriple-Phase-I$(s, p, o)$
19:     Let $r_{max} \coloneqq \arg\max_{r \in \{s,o\}} S(C(r))$, and let $r_{min}$ be the other vertex
20:     **if** $S(C(r_{max})) + |G^+(r_{min})| < (\alpha - 1)\frac{|G|}{n}$ **then**
21:        $S(C(r_{max})) \coloneqq S(C(r_{max})) + |G^+(r_{min})|$
22:        $S(C(r_{min})) \coloneqq S(C(r_{min})) - |G^+(r_{min})|$
23:        $C(r_{min}) \coloneqq C(r_{max})$

24: **function** AssignCommunities
25:     $N_k \coloneqq 0$ for each server $k \in \{1, \ldots, n\}$
26:     **for** each community $c$ occurring in the image of the mapping $C$ **do**
27:        $T(c) \coloneqq \arg\min_{k \in \{1,\ldots,n\}} |N_k|$
28:        $N_{T(c)} \coloneqq N_{T(c)} + S(c)$

29: **function** ProcessTriple-Phase-II$(s, p, o)$
30:     Add $(s, p, o)$ to $T(C(s))$

---

currently assigned to community $c$; rather, $S(c)$ provides us with the number of triples whose subject is assigned to community $c$. Because of that, $S(c_r)$ is initially set to $|G^+(r)|$ for each resource $r$, rather than to 1.

After initialisation, the algorithm processes each triple $\langle s, p, o \rangle \in G$ using function ProcessTriple-Phase-I. In line 19, the algorithm compares the sizes $S(C(s))$ and $S(C(o))$ of the communities to which $s$ and $o$, respectively, are currently assigned. It identifies $r_{max}$ as the resource whose current community size is larger, and $r_{min}$ as the resource whose current community size is smaller (ties are broken arbitrarily). The aim of this is to move $r_{min}$ into the community of $r_{max}$, but this is done only if, after the move, we can satisfy the requirement on the sizes of partition elements: if each community contains no more than $(\alpha - 1)\frac{|G|}{n}$ triples, we can later assign communities to servers greedily and the resulting partition elements will contain fewer than $\alpha\frac{|G|}{n}$ triples. This is reflected in the condition in line 19: if satisfied, the algorithm updates the sizes of the communities of $r_{max}$ and $r_{min}$ (lines 21–22), and it moves $r_{min}$ into the community of $r_{max}$ (line 23). If desired, $G$ can be processed repeatedly several times using function ProcessTriple-Phase-I to improve the communities.

Once all triples of $G$ are processed, function AssignCommunities assigns communities to servers. To this end, for each server $k$, the algorithm maintains the number $N_k$ of triples currently assigned to partition element $k$. Then, the communities from the image of $C$ (i.e., the communities that have 'survived' after shuffling the resources in the first phase) are assigned by greedily preferring

the least loaded server. Finally, using function PROCESSTRIPLE-PHASE-II, each triple $\langle s, p, o \rangle \in G$ is assigned to the server of community $C(s)$.

As in HDRF$_3$, our algorithm is not guaranteed to minimise the replication factor. However, the following result shows that the algorithm will honor the restriction on the sizes of partition elements for a suitable choice of $\alpha$.

**Proposition 2.** *Algorithm 2 produces a partition that satisfies $|G_i| \leq \alpha \frac{|G|}{n}$ for each $1 \leq i \leq n$ whenever $\alpha$ is selected such that*

$$\alpha > 1 + \frac{\max_r |G^+(r)|}{|G|}.$$

## 7    Evaluation

To see how partitioning affects distributed reasoning, we computed the materialisation for three large datasets, which we partitioned using subject hash partitioning (Hash), a variant of min-cut partitioning [22] (METIS), and our HDRF$_3$ and 2PS$_3$ algorithms. We introduce our datasets in Section 7.1; we present the test protocol in Section 7.2; and we discuss our results in Section 7.3.

### 7.1    Datasets

Apart from the well-known LUBM[3] benchmark, we are unaware of publicly available large RDF datasets that come equipped with complex datalog programs. Thus, we manually created programs for two well-known large datasets. All programs and datasets are available from the Web page from the introduction, and some statistical information about the datasets is shown in Table 1.

**LUBM-8K**  We used the LUBM dataset for 8,000 universities, containing 1.10 billion triples. Moreover, we used the *extended lower bound* datalog program by Motik et al. [20]. The program was constructed to stress-test reasoning systems, and it was obtained by translating the the OWL 2 RL portion of the LUBM ontology into datalog and manually adding several hard recursive rules that produce many redundant derivations. To the best of our knowledge, this program has not yet been used in the literature to test distributed RDF reasoners.

**WatDiv-1B**  The WatDiv[4] benchmark was developed as a test for SPARQL querying. We used the 1.09 billion triples provided by the creators of WatDiv. Since WatDiv does not include an ontology or datalog program, we manually produced a program consisting of 32 chain, cyclical, and recursive rules.

**MAKG**$^*$  The *Microsoft Academic Knowledge Graph* (MAKG) [7] is an RDF translation of the Microsoft Academic Graph—a heterogeneous dataset of scientific publication records, citations, authors, institutions, journals, conferences, and fields of study. The original MAKG dataset contains 8 billion triples and

---

[3] http://swat.cse.lehigh.edu/projects/lubm/
[4] https://dsg.uwaterloo.ca/watdiv/

**Table 1.** Datasets & Programs

| Dataset | Dataset Stats | | | Program Stats | | | Mat. Stats | | $\lambda$ |
|---|---|---|---|---|---|---|---|---|---|
| | triples (G) | res. (M) | deg. | rules | recr. | avg. body | triples (G) | der. (G) | |
| LUBM-8K | 1.10 | 260 | 4.21 | 103 | 3 | 1.20 | 2.66 | 63.45 | 819 |
| WatDiv-1B | 1.09 | 100 | 11.29 | 32 | 2 | 2.10 | 1.77 | 2.09 | 800 |
| MAKG* | 3.67 | 490 | 7.48 | 15 | 2 | 2.20 | 5.63 | 17.47 | 800 |

Legend: res. = #resources; deg. = triples/res.; recr. = #recursive rules; avg. body = average #body atoms; der. = #derivations; $\lambda$ = a HDRF$_3$ parameter

includes links to datasets in the Linked Open Data Cloud. To obtain a more manageable dataset, we selected a subset, which we call MAKG*, of 3.67 billion core triples. Since MAKG does not have an ontology, we manually created a datalog program consisting of 15 chain, cyclical, and recursive rules.

### 7.2   Test Protocol

As mentioned in Section 3, our DMAT system can be used with an arbitrary data partitioning strategy, so it provides us with an ideal testbed for our experiments. We ran our experiments on the Amazon EC2 cloud, with servers connected by 10 Gbps Ethernet. To compute the materialisation, we used ten servers of the r5 family, each equipped with a 2.3 GHz Intel Broadwell processor and 128 GB of RAM; the latter was needed since DMAT stores all data in RAM. We used an additional, smaller coordinator server to store the dictionary (i.e., mapping of resources to integers) and distribute the datalog program and the graphs to the cluster; this server did not participate in reasoning. Finally, we used another server with 784 GB of RAM to partition the data using METIS.

  To speed up loading times, we preprocessed all datasets by replacing all resources with integers. The coordinator distributed the triples to the workers for Hash, HDRF$_3$, and 2PS$_3$; for METIS, we loaded the precomputed partitions directly into the workers. To hash the triples' subjects, we simply multiplied the integer subject value by a large prime in order to randomise the distribution of the subjects. In our algorithms, we used $\alpha = 1.25$. With HDRF$_3$, we used $\delta = 0.25$ and we set $\lambda$ to the lowest value satisfying Proposition 1; the values of $\lambda$ thus vary for each dataset and are shown in Table 1. Finally, with 2PS$_3$, we processed the graphs twice in the first phase. After loading the dataset and the program into all servers, we computed the materialisation while recording the wall-clock time and the total number of messages sent between the servers.

### 7.3   Test Results & Discussion

For each of the four partitioning strategies, Table 2 shows the minimum, maximum, and median numbers of triples in partition elements, given as percentages of the overall numbers of triples. The table also shows the replication factor

(see Section 2 for a definition) and the time needed to compute the partitions. Finally, the table shows the reasoning times and the numbers of messages.

**Partition Times and Balance**  All partitioning schemes produced partition elements with sizes within the tolerance parameters: Hash achieves perfect balance if the hash function is uniform; METIS explicitly aims to equalise partition sizes; and our two algorithms do so by design and the choice of parameters. For all streaming methods, the partitioning times were not much higher than the time required to read the datasets from disk and send triples to their designated servers. In contrast, METIS partitioning took longer than materialisation on LUBM-8K and WatDiv-1B, and on MAKG* it ran out of memory even though we used a very large server equipped with 784 GB of RAM.

**Replication, Communication, and Reasoning Times**  Generally lowest replication factors were achieved with $2PS_3$: only METIS achieved a lower value on WatDiv-1B, and $HDRF_3$ achieved a comparable value on MAKG*. The replication factor of Hash was highest in all cases, closely followed by $HDRF_3$. Moreover, lower replication factors seem to corelate closely with decreased communication overhead; for example, the number of messages was significantly smaller on LUBM-8K and MAKG* with $2PS_3$ than with other schemes. This reduction seems to generally lead to a decrease in reasoning times: $2PS_3$ was the fastest than the other schemes on LUBM-8K and MAKG*; for the former, the improvement over Hash is by a factor of 2.25. However, the reasoning times do not always corelate with the replication factor: on WatDiv-1B, METIS and $2PS_3$ were slower than Hash and $HDRF_3$, despite exhibiting smaller replication factors.

**Workload Balance**  To investigate further, we show in Figure 1 the numbers of derivations and the total size of partial messages processed by each of the ten servers in the cluster. As one can see, partitioning the data into strongly connected clusters can introduce a workload imbalance: the numbers of derivations and messages per server are quite uniform for Hash and, to an extent, for $HDRF_3$; in contrast, with $2PS_3$ and METIS, certain servers seem to be doing much more work than others, particularly on WatDiv-1B and MAKG*. Thus, reducing communication seems to be important, but only to a point. For example, $2PS_3$ reduces communication drastically on LUBM-8K, and this seems to 'pay off' in terms of reasoning times. On MAKG*, the reduction in communication seems to lead to modest improvements in reasoning times, despite a more pronounced workload imbalance. On WatDiv-1B, however, communication overhead does not appear to be significant with any partitioning strategy, so the workload imbalance is the main determining factor of the reasoning times.

**Overall Performance**  In general, $2PS_3$ seems to provide a good performance mix: unlike METIS, it can be implemented without placing unrealistic requirements on the servers used for partitioning; it can significantly reduce communication; and, while this can increase reasoning times due to workload imbalances, such increases do not appear to be excessive. Thus, $2PS_3$ is a good alternative to hash partitioning, which has been the dominant technique used thus far.

**Table 2.** Partition & Reasoning

| Method | Partitioning Stats [n=10] | | | | | Reasoning Stats | |
|---|---|---|---|---|---|---|---|
| | Min (%) | Max (%) | Med (%) | RF | Time (s) | Time (s) | Messages (G) |
| LUBM-8K [1.10G triples] | | | | | | | |
| Hash | 10.00 | 10.00 | 10.00 | 1.60 | **530** | 17,400 | 71.67 |
| METIS | 9.24 | 10.66 | 9.98 | 1.19 | 15,300 | 12,580 | 15.44 |
| HDRF$_3$ | 9.35 | 10.47 | 10.00 | 1.43 | 590 | 15,740 | 46.05 |
| 2PS$_3$ | 9.06 | 10.35 | 10.00 | **1.08** | 700 | **7,740** | **9.22** |
| WatDiv-1B [1.09G triples] | | | | | | | |
| Hash | 10.00 | 10.00 | 10.00 | 2.48 | **520** | 1,870 | 8.95 |
| METIS | 9.70 | 10.35 | 10.00 | **2.16** | 15,100 | 2,690 | **4.54** |
| HDRF$_3$ | 10.00 | 10.00 | 10.00 | 2.48 | 590 | **1,850** | 8.95 |
| 2PS$_3$ | 9.92 | 10.02 | 10.00 | 2.40 | 1,080 | 2,520 | 8.81 |
| MAKG* [3.66G triples] | | | | | | | |
| Hash | 10.00 | 10.00 | 10.00 | 1.99 | **2,220** | 8,000 | 29.24 |
| METIS | Partitioning exhausted 784GB of memory | | | | | | |
| HDRF$_3$ | 10.00 | 10.00 | 10.00 | **1.66** | 3,500 | 7,160 | 26.15 |
| 2PS$_3$ | 9.91 | 10.06 | 10.00 | 1.67 | 3,640 | **6,870** | **24.70** |

## 8   Conclusion and Future Work

We have presented two novel algorithms for streaming partitioning of RDF data in distributed RDF systems. We have compared our methods against hashing and min-cut partitioning, which have been the dominant partitioning methods thus far. Our methods are much less resource-intensive than min-cut partitioning, and they are not significantly more complex than hashing. Particularly the 2PS$_3$ method often exhibits better reasoning performance, thus contributing to the scalability of distributed RDF systems. In our future work, we will aim to further improve the performance of reasoning by developing ways to reduce imbalances in the workload among servers. One possibility to achieve this might be to analyse the datalog program before partitioning and thus identify workload hotspots.

## Acknowledgments

This work was supported by the SIRIUS Centre for Scalable Access in the Oil and Gas Domain, and the EPSRC project AnaLOG.

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
