# OpenReview forum: "Streaming Partitioning of RDF Graphs for Datalog Reasoning"
_eswc-conferences.org/ESWC/2021/Conference/Research_Track — ESWC 2021 Research_

### Official Review · AnonReviewer2 · 2021-01-14
**Review:  Streaming Partitioning of RDF Graphs for Datalog Reasoning**

**Rating:** 2
**Confidence:** 4
**Impact:** 4
**Design And Technical Quality:** 4

**Review:**

The paper addresses the problem of RDF partitioning for distributed reasoning. Two new strategies are presented $HDRF_3$ and $2PS_3$, respectively based on $HDRF$ and $2PS$. The empirical evaluation includes hash and min-cut partitionings, and shows that these streaming graph partitioning algorithms can improve reasoning performance without unrealistic memory requirements.

**Comments:**

In the introduction, materialisation is mentioned as the approach often taken for reasoning. One or two sentences explaining the intuition for that choice should help to provide the motivation in the paper to improve materialisation instead of replacing it. Especially in the context of the datasets mentioned in section 7.1, with comparatively few rules.

Similarly, the motivation could use a deeper study of the behaviour of reasoning implementations in different contexts, e.g. expressiveness and relative size of ABox and TBox (or facts and rules for datalog). The 34 billion triples of UniProt would have a more meaningful contextualisation. This study is out of the scope of the current paper, but could impact it.

Minor comment: typo in “commpares”.


**Anonymity:**

Yes, I would like my review to remain anonymous.

**Reuse And Availability:**

3: Medium

**Strong Points:**

The paper discusses partitioning for distributed reasoning. The partitioning affects the reasoning for efficiency, with proxy metrics in partial messages and derivations. The efficiency is significant and even necessary for large graphs, e.g. METIS for MAKG.

While significant, the partitioning has no impact on the correctness of the reasoning, using the DMAT reasoner. They are both independent, which has a positive impact in:
1. The readability of the paper.
1. The applicability of the partitioning approaches.
1. The comparability of the partitioning approaches.


**Subreviewer:**

I submitted this review.

**Weak Points:**

The paper has a good motivation, that could possibly be presented more clearly: “Large graphs are necessary for certain use cases. Performance is crucial when reasoning over large graphs. The policies or algorithms used for graph partitioning may have a significant impact on reasoning performance, considering locality, load balance, and streaming partitioning”. The motivation (section 4) is in fact presented in the introduction, which could be shorter and more clear. For example, most of the contents in the paragraph about rule application might have been better in section 3 as related work.

The evaluation uses three datasets. Two had the datalog rules added for the evaluation, and LUBM-8K is constructed to stress-test reasoning systems. It is hard to know to what extent the results of the evaluation may be extrapolated to real use cases, which is always desirable although not always possible.

---

> ### Author Rebuttal · Authors · 2021-01-28
>
> Thank you for the review and for the helpful comments. We shall correct in the camera-ready version all typos mentioned and add further explanations (e.g., regarding materialisation, the need for large graphs, etc.).
>
> We also appreciate the comment about application to real use cases. This, in fact, was precisely why we developed the MAKG* test set: we wanted to validate our approach on data that is not automatically generated. Unfortunately, we just could not find a publicly available dataset that is both very large and also comes equipped with a reasonably complex set of rules. Therefore, we settled for using a real dataset, but generating the rules manually.

---

### Official Review · AnonReviewer1 · 2021-01-14
**Interesting paper but a bit more work is needed!**

**Rating:** 1
**Confidence:** 3
**Impact:** 4
**Design And Technical Quality:** 4

**Review:**

Comments after rebuttal:
I would like to thank the reviewers for their answers. Given they will update the CR version of the paper with the comments of all reviewers I believe it would be an interesting addition to the ESWC papers to be published.

==========

It is uncommon to include references in the abstract and I would suggest to remove them.

I would value a one sentence description of the main ideas behind HDRF and 2PS graph partitioning algorithms in the introduction. In addition, in the intro you say that a modification is required for those algorithms to work for RDF however but a sentence more here would intrigue more the reader.

In addition, I do not agree that in order to partition datasets using min-cut everything should be loaded into a single server. There are already distributed METIS implementations such as parMETIS etc. , streaming implementations such as SMP [x2] etc. Although you mention this in the sequel, “due to the high problem complexity, significant resources would still be required just to prepare the data” is not enough for appropriate argumentation.

However, I agree to that identifying how distributing RDF data affects distributed reasoning is still open to a large extent.

When adapting algorithms to the RDF model I would also expect special consideration for special edges coming such as subclassof, instanceof etc. or at least a discussion on these. In believe there are further chances of optimization besides ignoring labels in the edges and ensuring that triples with the same subject are assigned to the same server.

I do not understand the “streaming partitioning” at the title as in essence what you do is “distributed partitioning”. Although it is true that you adapt streaming graph partitioning methods, the “streaming” aspect should be better explained in the text.

It seems that in WatDiv Hash is better than your approach and is not appropriately explained why.

“2PS3 is a good alternative to hash partitioning, which has been the dominant technique used thus far.” In which cases? It seems that further investigation of the results should be performed, for better understanding when your approach is better.

Further, there have been paid already some attention in partitioning methods that might benefit your related work section [x3],[x4]

References
[x1] Karypis G. (2011) METIS and ParMETIS. In: Padua D. (eds) Encyclopedia of Parallel Computing. Springer, Boston, MA. https://doi.org/10.1007/978-0-387-09766-4_500

[x2]Ghizlane Echbarthi, Hamamache Kheddouci:Streaming METIS partitioning. ASONAM 2016: 17-24

[x3]Giannis Agathangelos, Georgia Troullinou, Haridimos Kondylakis , Kostas Stefanidis, Dimitris Plexousakis, Incremental Data Partitioning of RDF Data in SPARK, ESWC, pp. 50-54, 2018

 [x4]Giannis Agathangelos, Georgia Troullinou, Haridimos Kondylakis , Kostas Stefanidis, Dimitris Plexousakis, RDF Query Answering Using Apache Spark: Review and Assessment, ICDE, pp. 54-59, 2018


**Anonymity:**

Yes, I would like my review to remain anonymous.

**Reuse And Availability:**

4: High

**Strong Points:**

-The paper is easy to read and well structured, focusing on an important, open problem.

-The two algorithms proposed are rather interesting

-Datasets and implementation is available online



**Subreviewer:**

I submitted this review.

**Weak Points:**

I can easily see there is a room for improvement in terms of

a) adapting the proposed algorithms to the RDF setting

b) improvement on the reasoning time

c) better explaining the current observations (especially over WatDiv).

---

> ### Author Rebuttal · Authors · 2021-01-28
>
> Thank you for the review and for the helpful comments. We will update the paper with all suggestions mentioned in the review (e.g., references in the abstract, additional explanations, etc.).
>
> Regarding parallel and streaming METIS versions, thank you for pointing us to relevant related work we were not aware of before: we will add this to the camera-ready version. In our statement regarding the usage of resources, we simply observed that these techniques aim to solve an NP-hard problem, so such algorithms are likely to require significant resources. Table 2 provides some evidence for this: the partitioning times for METIS are orders of magnitude larger for LUBM-8K and WatDiv-1B, and moreover METIS could not partition MAKG*. It seems unlikely that using a distributed version of METIS would provide drastically different results; however, we agree that investigating this in practice would be very interesting.
>
> We will also consider and discuss in the camera-ready version all other references mentioned in the review.
>
> Thank you for the comment about special triples. Please note that the DMAT reasoner does not consider schema triples (e.g., rdfs:subClassOf) during reasoning: instead of using a fixed rule set, the schema triples in our approach are translated into a set of rules that are distributed to each node in the cluster. That is, <:A, rdfs:subClassOf, :B> gets converted to a rule <?X, rdf:type, :B> :- <?X, rdf:type, :A> . Because of that, we do not considered schema triples in our approach because these triples are not used during reasoning so their placement is not important. In contrast, rdf:type triples are taken into account during reasoning, but they are partitioned based on the subject, as one might expect. We fully appreciate that these issues should be clarified further, and we will do so in the camera-ready version.
>
> Regarding the use of the term "streaming", please note that this term generally does not imply that the input is processed only once; rather, thus usually means that the input is processed a fixed number of times. In fact, in their most general forms, both HDRF and 2PS use more than one pass through the input; however, both have been called "streaming" in the relevant literature. We simply follow the terminology used in the literature that we used as our starting point.
>
> The question regarding which partitioning scheme should be used for what kinds of datasets is a very difficult one, and similar questions arise often in practice. For example, in the area of description logic reasoning or first-order theorem proving, many different reasoning techniques have been developed, and it is often difficult/impossible to select a particular technique a priori. In such cases, the general approach seems to be to develop an ensemble of techniques, and to gather practical experience by applying the techniques in a range of different scenarios. We do agree, however, that further research in this area could be very interesting/useful.
>
> Regarding WatDiv, we said on page 13 that reasoning in this case does not seem to involve a lot of communication, so workload imbalance seems to be the dominant cost. We can think of clarifying this further if needed.

---

### Official Review · AnonReviewer5 · 2021-01-15
**The paper presents two algorithms that given an input RDF graph partition it into a set of graphs, which enable materialization of the input graph in a cloud.**

**Rating:** 1
**Confidence:** 4
**Impact:** 3
**Design And Technical Quality:** 5

**Review:**

Reasoning about very large graphs that cannot be loaded into memory might be problematic. The problem can be solved using distributed computing in the cloud. However, in this case, the original graphs must be stored in the cloud infrastructure and made available to each server. As it was shown in the previous work, standard data storing algorithms used in such frameworks as Hadoop or Spark might be inefficient. Therefore, the authors extend two streaming graph partitioning algorithms - HDRF and 2PS - which were developed for general (un)directed graphs, with features allowing the assignment of triples with the same subject to the same partition. This property might result in a better performance of distributed reasoning engines, such as DMAT, which was used in the evaluation. Results of the conducted experiments show that the agglomerative clustering method used in the $2PS_3$ algorithm was able to produce partitions resulting in shorter reasoning time for larger programs.

**Anonymity:**

Yes, I would like my review to remain anonymous.

**Reuse And Availability:**

5: Very High

**Strong Points:**

Overall, the paper is technically sound and the text is easy to read. The experiments were executed on really large graphs and definitely illustrate the benefits of using cloud computing for reasoning tasks.

There are only minor comments:
- Figure 7.3 -> Figure  1
- please highlight the best results in Table 3 in bold or italics
- personally, I would prefer $10^9$ and $10^6$ instead of G and M
- please specify the hashing algorithm you used in the evaluation
- the notion of "scale-free" graphs is used without a definition
- Algorithm 1, line 17: brackets might improve the readability -> $(R_k=0)$ ? 0  : ...

**Subreviewer:**

I submitted this review.

**Weak Points:**

The paper has some weak points in the evaluation, which are not discussed by the authors. As it appears the Hashing method had issues with large programs. However, if the program is small, the produced partitions allow a reasoner to compute the materialization faster than for $2PS_3$. Also, Hashing required less time for the computation of those partitions. In the case of WatDiv-1B, Hashing has the best partitioning time and the second-best reasoning time. Similarly, for the MAKG, Hashing required 10.22s for both partitioning and reasoning, whereas $2PS_3$ 10.51s. The total time cannot be ignored in the suggested scenario, since if only the reasoning time is important, e.g., a graph is partitioned once and then multiple queries are evaluated, then one might invest more time in finding a better partition. In addition, it would be interesting to see whether the performance remains stable when different programs are evaluated on the same partition. Evaluation of just one program does not allow one to judge the significance of obtained measurements, since there is a chance that some lucky guess of the partitioning algorithm allowed DMAT to find materialization faster.

---

> ### Author Rebuttal · Authors · 2021-01-28
>
> Thank you for the review and for the helpful comments. We will address all typos and comments mentioned in the camera-ready version, and we will clarify the term "scale-free".
>
> Regarding the question about the hashing function, we can certainly specify this. However, any hash function that distributes the results evenly should produce pretty much identical results.
>
> We do not quite understand the comments about large/small programs. In our experience, it is not the size of the program that matters, but rather the program's complexity. This, in turn, depends on (i) how hard it is to evaluate the bodies of rules, (ii) how many times a body is matched during reasoning, and (iii) whether the rules produce redundant derivations. All this has turned out to be quite hard to quantify by looking at the program in isolation.
>
> We also do not quite understand the comments about the total time. Table 2 shows that partitioning takes a similar amount of time for HDRF_3 and 2PS_3 on LUBM-8K and MAKG*, whereas on WatDiv HDRF_3 is is about 50% faster than 2PS_3. We can investigate and elaborate the latter point further if needed.
>
> The question of how to choose the partitioning method that works best for the given program seems to be partially related to the questions by AnonReviewer1 and AnonReviewer3, who wondered about how to choose the right approach for the graph at hand. We answered this question in our response to AnonReviewer1, and we hope that this partially also addresses this question. This question, however, introduces another interesting dimension: knowledge about the program might be used to optimise the partition. This was not the focus of our work thus far (i.e., our hope was to partition the data in a way that is agnostic w.r.t. the program), but this would be very interesting to consider in future research. In particular, the program might provide hints as to the type of joins that need to be optimised for, which could potentially lead to considerable performance gains.

---

### Official Review · AnonReviewer4 · 2021-01-15
**Interesting topic, well presented contribution**

**Rating:** 2
**Confidence:** 3
**Impact:** 3
**Design And Technical Quality:** 4

**Review:**


The paper addresses the challenge of RDF data partitioning for distributed reasoning. It presents two novel partitioning strategies for RDF graphs, both inspired and adapted by stream graph partition algorithms from recent literature.

The paper is very well presented and structure, and the motivation and challenges are clear. Given the importance of reasoning tasks and ever increasing size of RDF graphs, the contribution is also very relevant in the current days.

The original algorithms in which the work is based on are described. The adaptations to handle RDF graph are well justified and they take into consideration both the nature of RDF graphs and characteristics of reasoning tasks (e.g. subject-subject joins are very common). The authors also present the conditions in which balance requirements are guaranteed to hold, while their proof is given in a extended version of the paper, available online.

The experimental evaluation is clear and reproducible. Here I would have one remark: the reasoning tasks materialize triples which are then added back to the respective partition(s), based on their subject. Given that materializations can lead to a large number of new triples, which consequently affect the degree of the subjects, a question would be how the performance evolves over time, as more and more rules are executed and if there would be a point that a repartition would be required.

**Anonymity:**

Yes, I would like my review to remain anonymous.

**Reuse And Availability:**

4: High

**Strong Points:**

- Novel approaches for data partitioning to support distributed reasoning tasks
- Topic is very relevant and fitting to the conference
- Approach is well described and properly analyzed. Additional material, including proofs is available
- Reproducible

**Subreviewer:**

I submitted this review.

**Weak Points:**


- Partition performance is largely dependent on degrees of triples. Unclear how this is affected with the increasing number of materializations.

---

> ### Author Rebuttal · Authors · 2021-01-28
>
> Thank you for the review and for the helpful comments.
>
> We agree with the point that the derived triples might lead to load imbalances, and that repartitioning might be needed to address these. Nevertheless, we believe that keeping all triples with the same subject on a single machine would still be important: subject-subject joins are so common in RDF that, we believe, any other partitioning strategy would be vastly inferior. Thus, repartitioning might involve moving all triples with a particular subject to a different machine. This is an interesting problem that should be addressed in future research.

---

### Official Review · AnonReviewer3 · 2021-01-17
**A good comparison of RDF graph partitioning algorithms**

**Rating:** 2
**Confidence:** 4
**Impact:** 3
**Design And Technical Quality:** 4

**Review:**

Post rebuttal

Thank you for responding to my questions. Please include all the clarifications from the rebuttal in the final manuscript. In Figure 1, the labels for X and Y axes are missing. Placing the legend separately (outside of all the graphs) in one corner might help.

-----------------------------------------------------

Authors adapt two streaming graph partitioning algorithms, HDRF and 2PS, for RDF graphs. The major adaptation seems to be that the triples with the same subject are assigned to the same partition. These two techniques are compared with the hash and min-cut partitioning on datasets such as LUBM and WatDiv. The proposed techniques improve the datalog reasoning performance without consuming as much memory as the other two partitioning algorithms.

The existing partitioning approaches either consume a lot of memory (METIS) or create an imbalance (hash partitioning). So better RDF graph partitioning algorithms need to be investigated and the proposed algorithms is a step in this direction which seems to work fairly well. The work has been presented well and the evaluation is also good.

Since the major (only?) addition to the existing stream partitioning algorithms is to make sure that triples with the same subject are assigned to the same partition, it is not clear if any other changes were also required for the adaptation to RDF graphs. Apart from that, based on the evaluation, we cannot conclusively say that the proposed algorithms are better on all the datasets. In some cases, they are only better by only a small margin. But, nevertheless, this seems to be a promising direction to pursue further.

Other questions/comments
1) For a given RDF graph, how do you determine which partitioning technique would be the best? What aspects of the RDF graph influence the reasoning time? For example, 2PS3 performs well on LUBM and MAKG, but not on WatDiv.
2) Page 15, it looks like the label for the figure is wrong. There is no caption and the legends are missing.
3) Instead of DMAT, why not, for example, use RDFox on each server (after partitioning the graph)?
4) On all the three datasets, why were not the datalog equivalent of RDF/RDFS entailment rules used?
5) In Table 2, the units for triples is written as "G". Shouldn't it be billions?
6) What is the intuition/explanation for propositions 1 and 2? How did you get these expressions?
7) In the proposed two algorithms, since all the edges of the graph have to be traversed (a few times), doesn't this require loading of the entire graph into memory?
8) Page 6, Section 5.1, what is "scale-free"?
9) Two other relevant publications ([1], [2]) are missing.
10) Page 5, last line, the word communities is misspelt.
11) Page 8. It should be "encounter" rather than "encounters".
12) Page 9, Section 6. It should be "discuss" rather than "discusses".
13) Page 11, Section 7.1, "the" has been repeated twice.


[1] Grigoris Antoniou, Sotiris Batsakis, Raghava Mutharaju, Jeff Z. Pan, Guilin Qi, Ilias Tachmazidis, Jacopo Urbani, and Zhangquan Zhou. A Survey of Large-Scale Reasoning on the Web of Data. The Knowledge Engineering Review. 2018.
[2] Raghava Mutharaju, Sherif Sakr, Alessandra Sala, Pascal Hitzler. D-SPARQ: Distributed, Scalable and Efficient RDF Query Engine. Proceedings of the International Semantic Web Conference (ISWC 2013) Posters & Demonstrations Track, Sydney, Australia, October 23, 2013. CEUR Workshop Proceedings Vol. 1035, pp. 261-264.

**Anonymity:**

No, I would like my review to be deanonymized.

**Reuse And Availability:**

2: Low

**Strong Points:**

1) This is an interesting solution that can work in practice for a problem (partitioning approaches) that needs more investigation.
2) The paper is generally well-written.
3) Experiments are performed on very large RDF graphs (greater than 1 billion triples).

**Subreviewer:**

I submitted this review.

**Weak Points:**

1) The proposed streaming graph partitioning algorithms perform better comparatively on some of the datasets, but it is not clear that this would be the case for all the RDF graphs. It is also hard to say that they are clear winners.
2) Since the two proposed algorithms are adaptations of existing algorithms, the novelty aspect is not too high.

---

> ### Author Rebuttal · Authors · 2021-01-28
>
> Thank you for the review and for the helpful comments. We shall correct all typos, clarify what scale-free means, discuss all mentioned references, and add all clarifications requested. Our propositions have been proved by a series of equations that might be hard to explain intuitively, but we will see whether we can somehow summarise that.
>
> We do not quite understand the comment about Figure 1. There is a caption above the figure, and all subfigures have subcaptions too (e.g., LUBM-8K Partial Messages). Moreover, we have included the legend just once (in subfigure WatDiv-1B Derivations), but this legend applies to all figures equally. We did this to save space: including the legend in the other figures would cover a substantial portion of these. If this is not deemed sufficiently clear, we can perhaps see whether we can include a single legent beneath all figures.
>
> DMAT uses the triple storage, indexing, and retrieval mechanisms of RDFox. On top of that, DMAT implements specific algorithms for distributed reasoning, which deal with issues such communicating partial rule matches and distributing derived triples. We will clarify the relationship between DMAT and RDFox in the camera-ready version.
>
> Regarding the comment RDF/RDFS entailment, please note that none of the datasets we considered come equipped with an RDFS ontology. Moreover, RDFS reasoning is quite simple, and we wanted to test our approach with more complex rules.
>
> Regarding the comment about keeping the entire graph in memory, the main objective in streaming graph partitioning is to partition the graph while keeping in memory only a portion of the graph at a time. Our two algorithms follow this idea: they never keep all triples in memory at once (but they might make several passes through a graph). We can further clarify this if needed.
>
> Regarding the question of how to choose the right partitioning method, please refer to our answer to a similar question by reviewer AnonReviewer1.

---

### Decision · Program_Chairs · 2021-02-23

**Decision:**

Accept

**Comment:**

There is a large consensus among the reviewers that the paper tackles an important problem, proposes an interesting solution, and presents a convincing empirical evaluation at a large scale. They unanimously agree that the paper should be accepted. We encourage the authors to include the reviewers' comments in the camera-ready version of this paper. Moreover, we hope that the provided suggestions for future work will be pursued in other papers.